# Optically Active Telecom Defects in MoTe_2_ Fewlayers at Room Temperature

**DOI:** 10.3390/nano13091501

**Published:** 2023-04-27

**Authors:** Yuxin Lei, Qiaoling Lin, Sanshui Xiao, Juntao Li, Hanlin Fang

**Affiliations:** 1State Key Laboratory of Optoelectronic Materials and Technologies, School of Physics, Sun Yat-Sen University, Guangzhou 510275, China; 2Department of Electrical and Photonics Engineering, Technical University of Denmark, 2800 Kongens Lyngby, Denmark; 3NanoPhoton—Center for Nanophotonics, Technical University of Denmark, 2800 Kongens Lyngby, Denmark; 4Department of Microtechnology and Nanoscience (MC2), Chalmers University of Technology, 41296 Gothenburg, Sweden

**Keywords:** MoTe_2_, defect, optical fiber communication

## Abstract

The optical and electrical properties of semiconductors are strongly affected by defect states. The defects in molybdenum ditelluride (MoTe${}_{2}$) show the potential for quantum light emission at optical fiber communication bands. However, the observation of defect-related light emission is still limited to cryogenic temperatures. In this work, we demonstrate the deep defect states in MoTe${}_{2}$ fewlayers produced via a standard van der Waal material transfer method with a heating process, which enables light emission in the telecommunication O-band. The optical measurements show evidence of localized excitons and strong interaction among defects. Furthermore, the optical emission of defects depends on the thickness of the host materials. Our findings offer a new route for tailoring the optical properties of two-dimensional materials in optoelectronic applications.

## 1. Introduction

Two-dimensional (2D) semiconducting transition metal dichalcogenides (TMDs) hold great potential for applications in the field of optoelectronics and quantum technologies due to their large spin-orbit coupling [1], strong excitonic effect [2,3], and the crossover from indirect to direct bandgap [4,5]. The van der Waals structure allows for the engineering of artificial quantum materials for investigating novel physics by stacking various 2D materials and can be interfaced with quantum devices through integration with photonic structures [6]. The stacking of TMD monolayers into moiré superlattices allows for the electronic band engineering of a periodic 2D lattice of potential wells that facilitates the formation of identical quantum emitter arrays, by trapping the excitons within the potential wells [7,8]. Additionally, the excitons can be trapped by defects in 2D materials, which makes them behave as single photon emitters (SPEs). SPEs in TMDs were first observed at the edges of tungsten diselenide (WSe${}_{2}$) monolayers at temperatures below 30 K [9,10,11,12]. Reportedly, a higher working temperature up to ∼150 K could be achieved by improving the photoluminescence (PL) emission efficiency via defect density engineering and Purcell enhancement [13,14]. In stark contrast, the SPEs in hexagonal boron nitride (hBN) show a working temperature up to room temperature and even higher, which is attributed to the deep defect states that have large energy spacing with a conduction band minimum and a valence band maximum [15]. The semiconducting (2H phase) MoTe${}_{2}$ has a relatively small optical bandgap (∼1.1 eV) [16] and holds promise for integration with silicon photonics including the realization of photodetectors and lasers [17,18,19].

The bulk semiconducting MoTe${}_{2}$ has an indirect bandgap that becomes direct when the material is in the form of a monolayer [16]. Because the Mo-Te bonds are relatively weak, the thickness of the MoTe${}_{2}$ can be tailored layer by layer by laser etching [20]. Furthermore, the potential barrier between the semiconducting and the semimetallic (1T’) phase MoTe${}_{2}$ is smaller than other TMDs such as MoS${}_{2}$, WS${}_{2}$, MoSe${}_{2}$, and WSe${}_{2}$. The controllable phase transition from the 2H phase to the 1T’ phase could reduce the contact resistance. Such control over the phase of the material could be exploited in optoelectronic devices by using a 2H-MoTe${}_{2}$ as the channel and a 1T’-MoTe${}_{2}$ as the electrode [21].

The presence of defects affects both the electrical and optical properties of the MoTe${}_{2}$. The resulting mid-gap trap states hinder the transport of the photocurrent [22] of MoTe${}_{2}$-based FETs. On the other hand, the defects in the MoTe${}_{2}$ fewlayers can be activated by localized strain and form SPEs with emission wavelengths at optical fiber communication (OFC) bands at low temperatures [23]. To achieve the reliable performance of devices, understanding the impact of these defects is of significance. However, the optical properties of telecom defects in MoTe${}_{2}$ fewlayers at room temperature have remained unexplored.

In this work, we demonstrate the emergence of deep defect states in MoTe${}_{2}$ fewlayers assembled with the standard transfer technique with an additional heating process. In power-dependent and time-resolved PL measurements, we observed evidence of defect-bound excitons at room temperature. Additionally, their emission wavelengths were located in the fiberoptic O-band. The results could possibly offer a new path toward room-temperature quantum emitters for quantum information processing on a silicon platform and for light emission at OFC bands.

## 2. Materials and Methods

### 2.1. Sample Fabrication

Semiconducting MoTe${}_{2}$ and hBN flakes were mechanically exfoliated from bulk crystal (HQ Graphene) with Scotch tape and transferred to a Polydimethylsiloxane (PDMS) stamp by the previously reported dry-transfer method [24]. The fewlayer MoTe${}_{2}$ was identified by optical contrast and aligned with the bottom hBN flake with a home-built transfer setup. When the fewlayer MoTe${}_{2}$ was beside the bottom hBN flake, we heated the substrate to 100 ${}^{\circ }$C. The fewlayer MoTe${}_{2}$ made contact with the hBN flake due to the thermal expansion of the PDMS stamp. After the materials were in contact, and the temperature was stable, the heater was switched off, and the PDMS stamp gradually peeled off, leaving the fewlayer MoTe${}_{2}$ on top of the hBN flake. To avoid the oxidation of the MoTe${}_{2}$ flake as shown in a previous report [25], we transferred an additional hBN flake on top to form a sandwich structure that isolated the MoTe${}_{2}$ flake from the air (mainly oxygen and water).

### 2.2. Optical Measurements

To optically excite the MoTe${}_{2}$ flakes, a 532 nm continuous-wave (CW) laser was focused on the sample with a 50× objective (Olympus, LCPLN50XIR). The emitted light was collected using a multimode fiber and then sent to a near-IR spectrometer equipped with a nitrogen-cooled CCD. The sample was placed in a close-loop cryostat for temperature-dependent PL measurements. The time-resolved PL measurements were performed using a time-correlated single-photon counting technique with a time tagger. We excited the sample with a 640 nm pulsed laser (LDH-IB-640-B, PicoQuant) with a pulse width of <90 ps and a repetition rate of 40 MHz. A 900 nm longpass filter was used to block the excitation, and the signal with a wavelength longer than 1300 nm was sent to a single-photon detector (id220-FR, iDQ).

## 3. Results and Discussion

### 3.1. Generation of Deep Defect States

Figure 1a shows the fabrication process of the fewlayer MoTe${}_{2}$ samples introduced with defects. It should be emphasized that the sample was heated to 100 ${}^{\circ }$C when the MoTe${}_{2}$ was close to hBN, and the contact was achieved by the thermal expansion of the PDMS stamp. To prevent oxidation-induced device performance degradation, an additional hBN flake was transferred to the top of the MoTe${}_{2}$/hBN flake by fabrication without the heating process and formed a sandwich structure. Figure 1b shows an optical microscope image of Sample 1. The layer number distribution is marked by color-coded regions, which were identified by the optical contrast and PL emission wavelength. Under the optical pumping with a 532 nm CW laser, we observed two emission peaks P1 and P2 with an energy spacing of ∼117 meV (Figure 1c). The emission wavelength of P1 is associated with free A excitons in MoTe${}_{2}$ [16]. Regarding the origin of the P2 peak, we propose that the PL emission of MoTe${}_{2}$ flake is strongly mediated by the defect states generated during the transfer process and that the large energy spacing (i.e., deep defect states in the bandgap) enables the observation of the PL emission at room temperature. Note that the generation of a P2 peak was reproduced in Sample 2 (see Figure A1).

### 3.2. Characterizations of Defects

To confirm the presence of defect states in the MoTe${}_{2}$ flake, we performed power-dependent PL measurements. Figure 2a shows the PL spectrum of Sample 1 fitted to a bi-Gaussian function. The extracted integrated PL intensity of the two peaks as a function of the excitation power is plotted in Figure 2b. The P2 peak saturated with the increasing pump power, in good agreement with previously measured defect-bound excitons [11]. Furthermore, we found that the defects not only trapped the excitons but also affected the PL emission of the free excitons. With the increasing pump power, the emission photon energy of the P1 peak showed a clear blue shift (Figure 2d). We mainly attribute this energy shift to the enhanced repulsive exciton–exciton interaction [26] owing to the high density of defects. This interpretation is supported by the significant linewidth broadening of the P2 peak (Figure 2c).

Through the time-resolved PL dynamics measurements shown in Figure 3a, we measured the lifetime of the defect-bound excitons (i.e., P2). The extracted lifetime was about 115 ps, which was limited by the time resolution of our setup and was much shorter than the findings in previous works [9,11]. This can be attributed to the high density of the defects, which increased the nonradiative recombination rate. Interestingly, we found that both emission peaks showed a blue shift when decreasing the temperature from 297 K to 5 K, but the PL emission intensity and emission linewidth remained almost constant (Figure 3b). This behavior suggests the existence of strong phonon scattering, which can be caused by a high defect density. Further work is needed to obtain a deeper understanding of the physical mechanisms, which is beyond the scope of this work.

To further explore the mechanism of the generation of defects, we stacked the MoTe${}_{2}$/ hBN heterostructure on the SiO${}_{2}$/Si substrate without a heating process (Sample 3), as a reference sample. By heating the sample to 100 ${}^{\circ }$C under an ambient environment, we found that the PL emission diminished over time, and no additional peaks were found (Figure 3c and Figure A2), indicating that the generation of optically active defects was not related to heat-induced oxidization. Therefore, we conclude that the defects were generated due to the thermal expansion-induced strain from the PDMS stamp.

The optical properties of the host MoTe${}_{2}$ crystal depend on its thickness [16]. In the PL measurements of Sample 1, we found a clear redshift in the emission energy with the increased layer number (see Figure A3). Moreover, the increasing linewidth and energy shift in the emission line with the excitation power suggests the potential interactions in the defective Sample 1. We expect that the PL emission of defects was also modulated by the number of layers of the MoTe${}_{2}$ flake. The PL spectra of the MoTe${}_{2}$ fewlayers with different numbers of layers are shown in Figure 4a. The presence of defects was confirmed by the pump-power-dependent saturation behavior (Figure 4b). Compared to the PL emission of defects in the 2L-MoTe${}_{2}$, the defect emission increased for thicker layers. This emission intensity difference could be explained by two possible mechanisms: (1) the 2L-MoTe${}_{2}$ oxidized faster than the thicker layers [25], resulting in a weaker PL emission; (2) the thicker flakes were more mechanically stable [27] and gave rise to fewer defects generated during the assembly process, which would reduce the nonradiative recombinations. Similar to the 2L-MoTe${}_{2}$, the emission linewidth of the 4L- and 5L- MoTe${}_{2}$ showed a linewidth narrowing effect at pump powers above 1000 μW, as shown in Figure 4c. Figure 4d shows the continuous redshift in the defect emission from the 4L- and 5L-MoTe${}_{2}$, which mainly arose from the plasma-induced bandgap renormalization and the strong phonon scattering. Furthermore, the 1L- and 2L-MoTe${}_{2}$ were found to be direct band gap semiconductors, and the 3L-MoTe_2_ had an indirect bandgap [28]. The emission energy of the P2 peak of the 4L- and 5L-MoTe${}_{2}$ was almost the same at a relatively low excitation power (<2000 μW, see Figure 4d) and significantly lower than that of the 2L-MoTe${}_{2}$, which could possibly be attributed to the different band structures for these MoTe${}_{2}$ flakes.

## 4. Conclusions

In summary, we found the defect-mediated PL emission from the MoTe${}_{2}$ fewlayers, and the generation of defects was related to the transfer process. The short lifetime and the weak PL emission indicate a prominent nonradiative recombination process in our system. A new defect engineering technique such as the bombardment of a high-energy electron beam [14] is needed to control the defect density and study the physical mechanisms of defect–defect, defect–exciton, and defect-mediated exciton–exciton interactions contributing to the complex emission behavior. Moreover, reducing the defect density could possibly significantly enhance the PL emission [13], offering a new path towards quantum light sources at room temperature.

## Figures and Tables

**Figure 1 nanomaterials-13-01501-f001:**
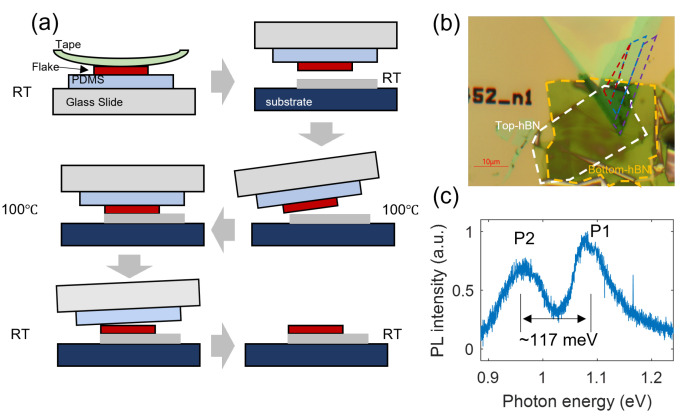
The generation of defect states in the MoTe${}_{2}$ flake. (**a**) Schematic of the sample fabrication process. (**b**) Bright-field microscope image of Sample 1. The color-coded regions represent the 2L-MoTe${}_{2}$ (purple), 4L-MoTe${}_{2}$ (blue), and 5L-MoTe${}_{2}$ (red). (**c**) PL emission of the hBN encapsulated 2L-MoTe${}_{2}$.

**Figure 2 nanomaterials-13-01501-f002:**
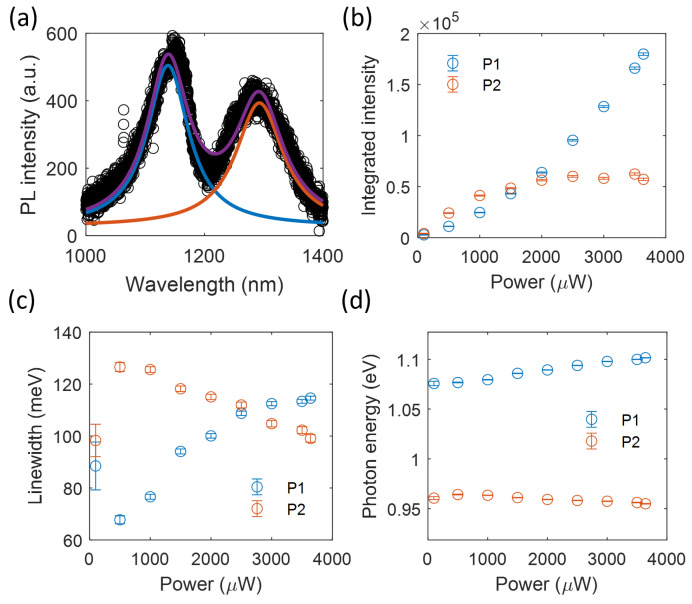
Power-dependent PL emission of the processed 2L-MoTe${}_{2}$. (**a**) Bi-Gaussian fit of the PL spectrum. Power-dependent integrated PL intensity (**b**), emission linewidth (**c**), and photon energy (**d**).

**Figure 3 nanomaterials-13-01501-f003:**
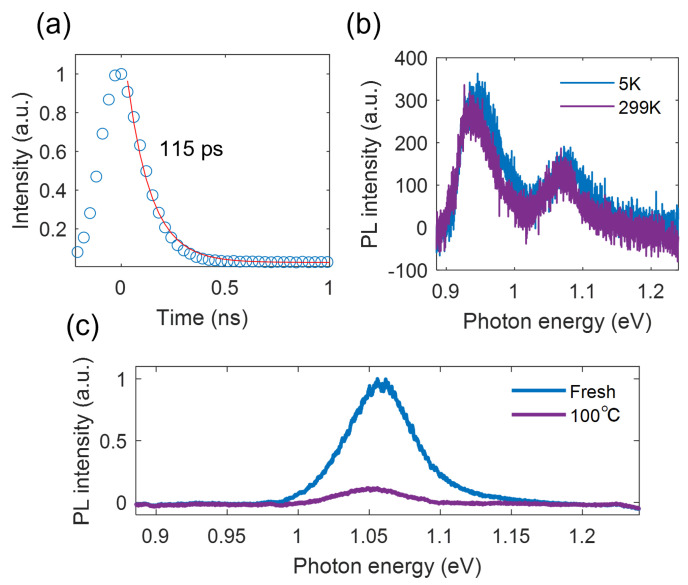
Features of the defect-bound excitons. (**a**) PL decay curve of the defects fitted with a single exponential function. (**b**) PL spectra at cryogenic and room temperatures. (**c**) The heating effect on the PL emission of the MoTe${}_{2}$ fewlayer of Sample 3.

**Figure 4 nanomaterials-13-01501-f004:**
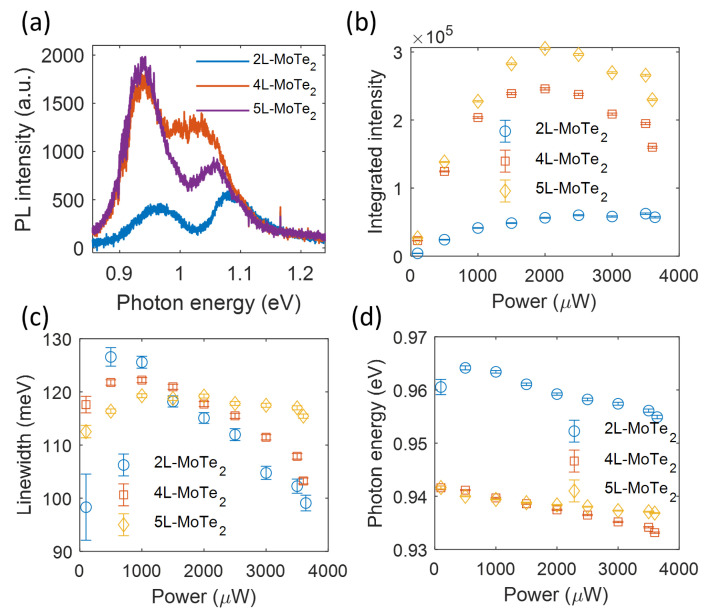
The layer-dependent PL emission of defects. (**a**) The layer-dependent PL spectrum under the same pump power. The power-dependent integrated PL intensity (**b**), emission linewidth (**c**), and emission photon energy (**d**) of the defect states in the 4L-MoTe${}_{2}$ and 5L-MoTe${}_{2}$.

## Data Availability

The data presented in this study are available on request from the corresponding authors.

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
