# Peer review of "Optically Active Telecom Defects in MoTe2 Fewlayers at Room Temperature"

_nanomaterials, 2023, doi:10.3390/nano13091501_

Round 1

Reviewer 1 Report

The authors report on PL emission from MoTe2 flakes obtained from a standard transfer technique with an additional heating process, they ascribe the PL emission to defects generated in the layers and investigate e.g. the power-dependent and layer-dependent emission spectra. While the discussion is somewhat lacking in depth, the research is timely and I can recommend the paper for publication, provided that the following comments are addressed:

1) Please check carefully for errors, such as "leaved" instead of "left" (p. 2, l. 53)

2) The authors mention that an additional hBN flake was transferred on top in order to avoid oxidation of the MoTe2 flake. How was the success of this approach determined? The authors should elaborate.

3) In the discussion, the authors state that "a new defect engineering technique will be needed to control the defect density" - some suggestions what that could be should be added.

Reviewer 2 Report

Review on "Optically active telecom-defects in MoTe 2 fewlayers at room temperature" submitted to nanomaterials by Yuxin Lei, Qiaoling Lin , Sanshui Xiao , Juntao Li and Hanlin Fang.

--

The authors report on the photoluminescence study on a defect related line in few hexagonal MoTe2. The samples are obtained by exfoliation and transfered between two hBN crystal. By engineering the transfer process (ie. tuning the temperature of the PDMS stamp while transfering to the lower hBN surface), the authors reports the apparition of a stable defect band below the band gap of hexagonal MoTe2.

I recommend a major revision so that the authors take the time to rework their methods and figures. Although I would tend to agree with main claim, the overall presentation is confusing. The writing of last discussion (line 110+) is not clear.

See the remarks detailed below:

---General----

* MoTe2 is well known for phase switching between semiconducting hexagonal and gap-less 1T' / 1Td. While it may be obvious to the authors, they should specify that their work relates to hexagonal MoTe2 (in the introduction paragraph near line 30).

* hexagonal MoTe2 shows the direct to indirect band gap transition with thickness which is iconic to many TMD. This is absent in the introduction text (we have to wait until line 110, with no proper reference for this particular sentence). However the authors cite the relevant reference (ref[16]) in the introduction.

==> Such basic concepts, studied MoTe2 being hexagonal, with expected direct to indirect band gap transition. should be performed in the introduction. This will help situating the authors works in the field.

----Methods----------

* The x-axis of all PL is in nanometer. While this is technically correct, most of the literature uses eV to characterize the defect line and band gap.

However, FWHM in "nm", fitted from Gaussian in "nm" points towards a non-conventional physic model that requires clarification.

* The dual meV/nm use in Fig1.c is only adding to the confusion.

==> please change the "nanometer" x-axis to "eV" in all PL result so we can compare with the literature

==> please fit all spectra in "eV" to extract the correct FWHM in the energy

[I agree that optical fiber specialists refer to silica transparency windows in "nanometers" or "microns" but this is not point here]

----Critical point ---

* Fig 3c : The spectra should be normalized for the reader to appreciate the presence/absence of shift in the PL lines.

* The Fig3c should compare the following normalized spectra the data from 1a.

"100°C transfer"

"RoomTemp transfer"

"RoomTemp transfer annealed at 100°C"

=> the comparison between these 3 (normalized) spectra is the main claim of the manuscript. These should be compared at length and in details in the main text.

* Possibly add a fig3d with the same 3 spectra without normalization to be further discussed in the main text.

---Other (not critical)----

* Fig 3a : Time resolved PL is only shown for the "defect line". Do the authors have data on the time resolved PL on the "main line" ?

------- Presentation------

* Fig 4, please add the 2L-MoTe2 data in (b)(c)(d) so that the readers can appreciate the difference (which are detailed in the main text).

[please also correct the reversal of "linewidth" "wavelength" subplots between fig2(c-d) and fig4(c-d), harder for the reader to compare]

* Fig4 should also present the associated shift in PL energy with increasing layer thickness for P1, so the reader can evaluate the respective shift of P1 vs P2. Possibly add this in SI if it obfuscates fig4 too much.

*Follow up of previous point : How does the defect line energy changes with sample thickness ? does it follow the same variation as the X0 line ? Is is compatible with a deep trap level at fixed energy ?

------Conclusion ---------------

* I do not understand the reasoning of the authors in line 114-115.

Fact : Thicker crystal present increased defect related emission (fig4a)

[line 117] "thicker flakes are more mechanically stable" => should give less "defect related signal", not more

at this point of the manuscript, the authors seem to use "defect" both for the studied "radiative defect line" and for other "non-radiative defect".

=> please clarify what type of defect is doing what and their respective origin.

Reviewer 3 Report

The main purpose of this article is to study optically active defects in MoTe2. The Authors demonstrate the generation of deep defect states in MoTe2 few-layers via a standard van der Waal material transfer method with a heating process, which enables the observation of light emission in the telecommunication O-band. The short lifetime with weak PL emission indicates the prominent non-radiative recombination process.  It was shown presence of localized excitons with the short lifetime, that indicates the strong interaction among defects. Optical emission behavior of defects depends on the thickness of the host materials. This allows tailoring the optical properties of two-dimensional materials in optoelectronic applications.

The strength of the work is that the obtained results could possibly offer a new avenue for room-temperature quantum emitters toward the quantum information process on a silicon platform and the band structure engineering for the light emission at OFC bands.

Work seems interesting. In addition to a few general comments, I have no objections to the substantive side

-Point 3 "results" should be changed to "results and discussion"

- Point 4 "discussion" should be changed to "conclusions"

- Literature was cited mainly from the 2014 - 2018 period, if possible, please analyze the latest literature.

- English can be improved

-Literature in the text is disordered, e.g. [24] occurs directly after [20]

Reviewer 4 Report

The manuscript presents a spectroscopic PL investigation of optically-active defects generated at room temperature in MoTe2 via a transfer method involving an heating step. I believe the results presented in the paper are consistent and interesting for publication in Nanomaterials.

Despite I understand this was not the goal of the present publication, I encourage the authors to at least formulate some hypothesis on what kind of defects are actually generated (point defects, edge defects?). This could pave the way to a future morphological characterization of such defects.
